# High-Fidelity Audio Compression
# with Improved RVQGAN

**Rithesh Kumar***
Descript, Inc.

**Prem Seetharaman***
Descript, Inc.

**Alejandro Luebs**
Descript, Inc.

**Ishaan Kumar**
Descript, Inc.

**Kundan Kumar**
Descript, Inc.

## Abstract

Language models have been successfully used to model natural signals, such as images, speech, and music. A key component of these models is a high quality neural compression model that can compress high-dimensional natural signals into lower dimensional discrete tokens. To that end, we introduce a high-fidelity universal neural audio compression algorithm that achieves 90x compression of 44.1 KHz audio into tokens at just 8kbps bandwidth. We achieve this by combining advances in high-fidelity audio generation with better vector quantization techniques from the image domain, along with improved adversarial and reconstruction losses. We compress all domains (speech, environment, music, etc.) with a single universal model, making it widely applicable to generative modeling of all audio. We compare with competing audio compression algorithms, and find our method outperforms them significantly. We provide thorough ablations for every design choice, as well as open-source code and trained model weights. We hope our work can lay the foundation for the next generation of high-fidelity audio modeling.

## 1 Introduction

Generative modeling of high-resolution audio is difficult due to high dimensionality (~44,100 samples per second of audio) [24, 19], and presence of structure at different time-scales with both short and long-term dependencies. To mitigate this problem, audio generation is typically divided into two stages: 1) predicting audio conditioned on some intermediate representation such as mel-spectrograms [24, 28, 19, 30] and 2) predicting the intermediate representation given some conditioning information, such as text [35, 34]. This can be interpreted as a hierarchical generative model, with observed intermediate variables. Naturally, an alternate formulation is to learn the intermediate variables using the variational auto-encoder (VAE) framework, with a learned conditional prior to predict the latent variables given some conditioning. This formulation, with continuous latent variables and training an expressive prior using normalizing flows has been quite successful for speech synthesis [17, 36].

A closely related idea is to train the same varitional-autoencoder with discrete latent variables using VQ-VAE [38]. Arguably, discrete latent variables are a better choice since expressive priors can be trained using powerful autoregressive models that have been developed for modeling distributions over discrete variables [27]. Specifically, transformer language models [39] have already exhibited the capacity to scale with data and model capacity to learn arbitrarily complex distributions such as text[6], images[12, 44], audio [5, 41], music [1], etc. While modeling the prior is straightforward, modeling the discrete latent codes using a quantized auto-encoder remains a challenge.

---

*Equal contribution to this work. Address correspondence to papers@descript.com, or raise an issue at `https://github.com/descriptinc/descript-audio-codec`.

37th Conference on Neural Information Processing Systems (NeurIPS 2023).

Learning these discrete codes can be interpreted as a lossy compression task, where the audio signal is compressed into a discrete latent space by vector-quantizing the representations of an autoencoder using a fixed length codebook. This audio compression model needs to satisfy the following properties: 1) Reconstruct audio with high fidelity and free of artifacts 2) Achieve high level of compression along with temporal downscaling to learn a compact representation that discards low-level imperceptible details while preserving high-level structure [38, 33] 3) Handle all types of audio such as speech, music, environmental sounds, different audio encodings (such as mp3) as well as different sampling rates using a single universal model.

While the recent neural audio compression algorithms such as SoundStream [46] and EnCodec [8] partially satisfy these properties, they often suffer from the same issues that plague GAN-based generation models. Specifically, such models exhibit audio artifacts such as tonal artifacts [29], pitch and periodicity artifacts [25] and imperfectly model high-frequencies leading to audio that are clearly distinguishable from originals. These models are often tailored to a specific type of audio signal such as speech or music and struggle to model generic sounds. We make the following contributions:

- We introduce **Improved RVQGAN** a high fidelity universal audio compression model, that can compress 44.1 KHz audio into discrete codes at 8 kbps bitrate (~90x compression) with minimal loss in quality and fewer artifacts. Our model outperforms state-of-the-art methods by a large margin even at lower bitrates (higher compression) , when evaluated with both quantitative metrics and qualitative listening tests.
- We identify a critical issue in existing models which don't utilize the full bandwidth due to **codebook collapse** (where a fraction of the codes are unused) and fix it using improved codebook learning techniques.
- We identify a side-effect of **quantizer dropout** - a technique designed to allow a single model to support variable bitrates, actually hurts the full-bandwidth audio quality and propose a solution to mitigate it.
- We make impactful design changes to existing neural audio codecs by adding periodic inductive biases, multi-scale STFT discriminator, multi-scale mel loss and provide thorough ablations and intuitions to motivate them.
- Our proposed method is a universal audio compression model, capable of handling speech, music, environmental sounds, different sampling rates and audio encoding formats.

We provide code [1], models, and audio samples [2] that we encourage the reader to listen to.

## 2 Related Work

**High fidelity neural audio synthesis**: Recently, generative adversarial networks (GANs) have emerged as a solution to generate high-quality audio with fast inference speeds, due to the feed-forward (parallel) generator. MelGAN [19] successfully trains a GAN-based spectrogram inversion (neural vocoding) model. It introduces a multi-scale waveform discriminator (MSD) to penalize structure at different audio resolutions and a feature matching loss that minimizes L1 distance between discriminator feature maps of real and synthetic audio. HifiGAN [18] refines this recipe by introducing a multi-period waveform discriminator (MPD) for high fidelity synthesis, and adding an auxiliary mel-reconstruction loss for fast training. UnivNet [16] introduces a multi-resolution spectrogram discriminator (MRSD) to generate audio with sharp spectrograms. BigVGAN [21] extends the HifiGAN recipe by introducing a periodic inductive bias using the Snake activation function [47]. It also replaces the MSD in HifiGAN with the MRSD to improve audio quality and reduce pitch, periodicity artifacts [25]. While these the GAN-based learning techniques are used for vocoding, these recipes are readily applicable to neural audio compression. Our Improved RVQGAN model closely follows the BigVGAN training recipe, with a few key changes. Our model uses a new multi-band multi-scale STFT discriminator that alleviates aliasing artifacts, and a multi-scale mel-reconstruction loss that better model quick transients.

**Neural audio compression models**: VQ-VAEs [38] have been the dominant paradigm to train neural audio codecs. The first VQ-VAE based speech codec was proposed in [13] operating at 1.6 kbps. This model used the original architecture from [38] with a convolutional encoder and an autoregressive

---

[1]https://github.com/descriptinc/descript-audio-codec
[2]https://descript.notion.site/Descript-Audio-Codec-11389fce0ce2419891d6591a68f814d5

wavenet [27] decoder. SoundStream [46] is one of the first universal compression models capable of handling diverse audio types, while supporting varying bitrates using a single model. They use a fully causal convolutional encoder and decoder network, and perform residual vector quantization (RVQ). The model is trained using the VQ-GAN [12] formulation, by adding adversarial and feature-matching losses along with the multi-scale spectral reconstruction loss. EnCodec [8] closely follows the SoundStream recipe, with a few modifications that lead to improved quality. EnCodec uses a multi-scale STFT discriminator with a multi-scale spectral reconstruction loss. They use a loss balancer which adjusts loss weights based on the varying scale of gradients coming from the discriminator.

Our proposed method also uses a convolutional encoder-decoder architecture, residual vector quantization and adversarial, perceptual losses. However, our recipe has the following key differences: 1) We introduce a periodic inductive bias using Snake activations [47, 21] 2) We improve codebook learning by projecting the encodings into a low-dimensional space [44] 3) We obtain a stable training recipe using best practices for adversarial and perceptual loss design, with fixed loss weights and without requiring a sophisticated loss balancer. We find that our changes lead to a near-optimal effective bandwidth usage. This allows our model to outperform EnCodec even with 3x lower bitrate.

**Language modeling of natural signals** : Neural language models have demonstrated great success in diverse tasks such as open-ended text generation [6] with in-context learning capabilities. A key-component of these models is self-attention [39], which is capable of modeling complex and long-range dependencies but suffers from a quadratic computational cost with the length of the sequence. This cost is unacceptable for natural signals such as images and audio with very high dimensionality, requiring a compact mapping into a discrete representation space. This mapping is typically learnt using VQ-GANs [12, 44], followed by training an autoregressive Transformer on the discrete tokens. This approach has shown success across image [45, 44, 32], audio [5, 41], video and music [9, 1] domains. Codecs like SoundStream and EnCodec have already been used in generative audio models, like AudioLM [5], MusicLM [1], and VALL-E [41]. Our proposed model can serve as a drop-in replacement for the audio tokenization model used in these methods, allowing for highly superior audio fidelity, and more efficient learning due to our maximum entropy code representation.

## 3 The Improved RVQGAN Model

Our model is built on the framework of VQ-GANs, following the same pattern as SoundStream [46] and EnCodec [8]. Our model uses the fully convolutional encoder-decoder network from SoundStream, that performs temporal downscaling with a chosen striding factor. Following recent literature, we quantize the encodings using Residual Vector Quantization (RVQ), a method that recursively quantizes residuals following an initial quantization step with a distinct codebook. Quantizer dropout is applied during training to enable a single model that can operate at several target bitrates. Our model is similarly trained using a frequency domain reconstruction loss along with adversarial and perceptual losses.

| Codec | Sampling rate (kHz) | Target bitrate (kbps) | Striding factor | Frame rate (Hz) | # of 10-bit codebooks | Compression factor |
|---|---|---|---|---|---|---|
| Proposed | 44.1 | 8 | 512 | 86 | 9 | 91.16 |
| EnCodec | 24 | 24 | 320 | 75 | 32 | 16 |
| | 48 | 24 | 320 | 150 | 16 | 32 |
| SoundStream | 24 | 6 | 320 | 75 | 8 | 64 |

Table 1: Comparison of compression approaches.

An audio signal with sampling rate $f_s$ (Hz), encoder striding factor $M$, and $N_q$ layers of RVQ produce a discrete code matrix of shape $S \times N_q$, where $S$ is the frame rate defined as $f_s/M$. Table 1 compares our proposed model against baselines to contrast the compression factors and the frame rate of latent codes. Note that the target bitrate mentioned is an upper bound, since all models support variable bitrates. Our model achieves a higher compression factor compared to all baseline methods while outperforming them in audio quality, as we show later. Finally, a lower frame rate is desirable when training a language model on the discrete codes, as it results in shorter sequences.

### 3.1 Periodic activation function

Audio waveforms are known to exhibit high periodicity (especially in voiced components, music, etc.) While current non-autoregressive audio generation architectures are capable of generating high fidelity audio, they often exhibit jarring pitch and periodicity artifacts [25]. Moreover, common

neural network activations (such as Leaky ReLUs) are known to struggle with extrapolating periodic signals, and exhibit poor out-of-distribution generalization for audio synthesis [21].

To add a periodic inductive bias to the generator, we adopt the Snake activation function proposed by Liu et al. [47] and introduced to the audio domain in the BigVGAN neural vocoding model [21]. It is defined as $\text{snake}(x) = x + \frac{1}{\alpha}\sin^2(\alpha x)$, where $\alpha$ controls the frequency of periodic component of the signal. In our experiments, we find replacing Leaky ReLU activations with Snake function to be an influential change that significantly improves audio fidelity (Table 2).

## 3.2 Improved residual vector quantization

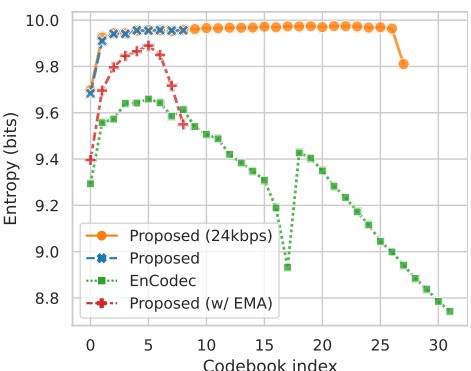

Figure 1: Entropy for each codebook, computed using code usage statistics across a large test set.

While vector quantization (VQ) is a popular method to train discrete auto-encoder, there are many practical struggles when training them. Vanilla VQ-VAEs struggle from low codebook usage due to poor initialization, leading to a significant portion of the codebook being unused. This reduction in effective codebook size leads to an implicit reduction in target bitrate, which translates to poor reconstruction quality.

To mitigate this, recent audio codec methods use k-means clustering to initialize the codebook vectors, and manually employ randomized restarts [9] when certain codebooks are unused for several batches. However, we find that the EnCodec model trained at 24kbps target bitrate, as well as our proposed model with the same codebook learning method (Proposed w/ EMA) still suffers from codebook under-utilization (Figure 1).

To address this issue, we use two key techniques introduced in the Improved VQGAN image model[44] to improve codebook usage: factorized codes and L2-normalized codes. Factorization decouples code lookup and code embedding, by performing code lookup in a low-dimensional space (8d or 32d) whereas the code embedding resides in a high dimensional space (1024d). Intuitively, this can be interpreted as a code lookup using only the principal components of the input vector that maximally explain the variance in the data. The L2-normalization of the encoded and codebook vectors converts euclidean distance to cosine similarity, which is helpful for stability and quality [44].

These two tricks along with the overall model recipe significantly improve codebook usage, and therefore bitrate efficiency (Figure 1) and reconstruction quality (Table 2), while being simpler to implement. Our model can be trained using the original VQ-VAE codebook and commitment losses [38], without k-means initialization or random restarts. The equations for the modified codebook learning procedure are written in Appendix A

## 3.3 Quantizer dropout rate

Quantizer dropout was introduced in SoundStream [46] to train a single compression model with variable bitrate. The number of quantizers $N_q$ determine the bitrate, so for each input example we randomly sample $n \sim \{1, 2, \ldots, N_q\}$ and only use the first $n_q$ quantizers while training. However, we noticed that applying quantizer dropout degrades the audio reconstruction quality at full bandwidth (Figure 2).

To address this problem, we instead apply quantizer dropout to each input example with some probability $p$. Interestingly, we find that dropout probability $p = 0.5$ closely matches the reconstruction quality of baseline at lower bitrates, while closing the gap to full-bandwidth quality of a model trained without quantizer dropout ($p = 0.0$).

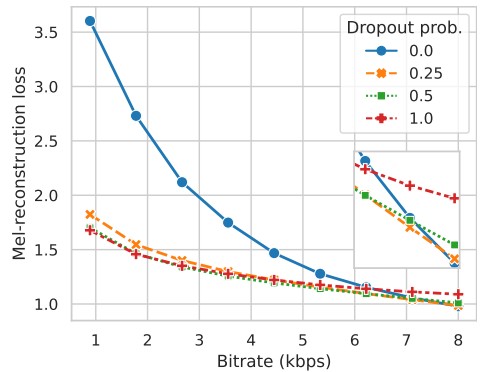

Figure 2: Effect of quantizer dropout on audio quality vs bitrate.

Moreover, we provide additional insight into the practical behavior of quantizer dropout and it's interaction with RVQ. Firstly, we find that these techniques put together lead the quantized codes to learn most-significant to least significant bits of information with each additional quantizer. When the codes are reconstructed with $1 \ldots N_q$ codebooks, we can see each codebook adds increasing amounts of fine-scale detail. We believe this interaction is beneficial when training hierarchical generative models on top of these codes [5, 41, 1], for example to partition the codes into "coarse" tokens (denoting the most significant codes) and "fine" tokens.

### 3.4 Discriminator design

Like prior work, we use multi-scale (MSD) and multi-period waveform discriminators (MPD) which lead to improved audio fidelity. However, spectrograms of generated audio can still appear blurry, exhibiting over-smoothing artifacts in high frequencies[16]. The multi-resolution spectrogram discriminator (MRSD) was proposed in UnivNet to fix these artifacts and BigVGAN [21] found that it also helps to reduce pitch and periodicity artifacts. However, using magnitude spectrograms discards phase information which could've been otherwise utilized by the discriminator to penalize phase modeling errors. Moreover, we find that high-frequency modeling is still challenging for these models especially at high sampling rates.

To address these issues, we use a complex STFT discriminator [46] at multiple time-scales [8] and find that it works better in practice and leads to improved phase modeling. Additionally we find that splitting the STFT into sub-bands slightly improves high frequency prediction and mitigates aliasing artifacts, since the discriminator can learn discriminative features about a specific sub-band and provide a stronger gradient signal to the generator. Multi-band processing was earlier proposed in [43] to predict audio in sub-bands which are subsequently summed to produce the full-band audio.

### 3.5 Loss functions

**Frequency domain reconstruction loss:** while the mel-reconstruction loss [18] is known to improve stability, fidelity and convergence speed, the multi-scale spectral losses[42, 11, 15] encourage modeling of frequencies in multiple time-scales. In our model, we combine both methods by using a L1 loss on mel-spectrograms computed with window lengths of $[32, 64, 128, 256, 512, 1024, 2048]$ and hop length set to window_length / 4. We especially find that using the lowest hop size of 8 improves modeling of very quick transients that are especially common in the music domain.

EnCodec [8] uses a similar loss formulation, but with both L1 and L2 loss terms, and a fixed mel bin size of 64. We find that fixing mel bin size leads to holes in the spectrogram especially at low filter lengths. Therefore, we use mel bin sizes $[5, 10, 20, 40, 80, 160, 320]$ corresponding to the above filter lengths which were verified to be correct by manual inspection.

**Adversarial loss:** our model uses the multi-period discriminator [18] for waveform discrimination, as well as the proposed multi-band multi-scale STFT discriminator for the frequency domain. We use the HingeGAN [22] adversarial loss formulation, and apply the L1 feature matching loss [19].

**Codebook learning:** we use the simple codebook and commitment losses with stop-gradients from the original VQ-VAE formulation [38], and backpropagate gradients through the codebook lookup using the straight-through estimator [3].

**Loss weighting:** we use the loss weightings of $15.0$ for the multi-scale mel loss, $2.0$ for the feature matching loss, $1.0$ for the adversarial loss and $1.0, 0.25$ for the codebook and commitment losses respectively. These loss weightings are in line with recent works [18, 21] (which use $45.0$ weighting for the mel loss), but simply rescaled to account for the multiple scales and $\log_{10}$ base we used for computing the mel loss. We don't use a loss balancer as proposed in EnCodec [8].

## 4 Experiments

### 4.1 Data sources

We train our model on a large dataset compiled of speech, music, and environmental sounds. For speech, we use the DAPS dataset [26], the clean speech segments from DNS Challenge 4 [10], the Common Voice dataset [2], and the VCTK dataset [40]. For music, we use the MUSDB dataset

[31], and the Jamendo dataset [4]. Finally, for environmental sound, we use both the balanced and unbalanced train segments from AudioSet [14]. All audio is resampled to 44kHz.

During training, we extract short excerpts from each audio file, and normalize them to -24 dB LUFS. The only data augmentation we apply is to randomly shift the phase of the excerpt, uniformly. For evaluation, we use the evaluation segments from AudioSet [14], two speakers that are held out from DAPS [26] (F10, M10) for speech, and the test split of MUSDB [31]. We extract 3000 10-second segments (1000 from each domain), as our test set.

## 4.2 Balanced data sampling

We take special care in how we sample from our dataset. Though our dataset is resampled to 44kHz, the data within it may be band-limited in some way. That is, some audio may have had an original sampling rate much lower than 44kHz. This is especially prevalent in speech data, where the true sampling rates of the underlying data can vary greatly (e.g. the Common Voice data is commonly 8-16kHz). When we trained models on varying sampling rates, we found that the resultant model often would not reconstruct data above a certain frequency. When investigating, we found that this threshold frequency corresponded to the average true sampling rate of our dataset. To fix this, we introduce a *balanced data sampling* technique.

We first split our dataset into data sources that we know to be *full-band* - they are confirmed to contain energy in frequencies up to the desired Nyquist frequency (22.05kHz) of the codec - and data sources where we have no assurances of the max frequency. When sampling batches, we make sure that a full-band item is sampled. Finally, we ensure that in each batch, there are an equal number of items from each domain: speech, music, and environmental sound. In our ablation study, we examine how this balanced sampling technique affects model performance.

## 4.3 Model and training recipe

Our model consists of a convolutional encoder, a residual vector quantizer, and a convolutional decoder. The basic building block of our network is a convolutional layer which either upsamples or downsamples with some stride, followed by a residual layer consisting of convolutional layers interleaved with non-linear Snake activations. Our encoder has 4 of these layers, each of which downsamples the input audio waveform at rates $[2, 4, 8, 8]$. Our decoder has 4 corresponding layers, which upsample at rates $[8, 8, 4, 2]$. We set the decoder dimension to $1536$. In total, our model has 76M parameters, with 22M in the encoder, and 54M in the decoder. We also examine decoder dimensions of 512 (31M parameters) and 1024 (49M parameters).

We use the multi-period discriminator [18], and a complex multi-scale STFT discriminator. For the first, we use periods of $[2, 3, 5, 7, 11]$, and for the second, we use window lengths $[2048, 1024, 512]$, with a hop-length that is $1/4$ the window length. For band-splitting of the STFT, we use the band-limits $[0.0, 0.1, 0.25, 0.5, 0.75, 1.0]$. For the reconstruction loss, we use distance between log-mel spectrograms with window lengths $[32, 64, 128, 256, 512, 1024, 2048]$, with corresponding number of mels for each of $[5, 10, 20, 40, 80, 160, 320]$. The hop length is $1/4$ of the window length. We use feature matching and codebook losses, as described in Section 3.5.

For our ablation study, we train each model with a batch size of 12 for 250k iterations. In practice, this takes about 30 hours to train on a single GPU. For our final model, we train with a batch size of 72 for 400k iterations. We train with excerpts of duration 0.38s. We use the AdamW optimizer [23] with a learning rate of $1e - 4$, $\beta_1 = 0.8$, and $\beta_2 = 0.9$, for both the generator and the discriminator. We decay the learning rate at every step, with $\gamma = 0.999996$.

## 4.4 Objective and subjective metrics

To evaluate our models, we use the following objective metrics:

1. ViSQOL [7]: an intrusive perceptual quality metric that uses spectral similarity to the ground truth to estimate a mean opinion score.
2. Mel distance: distance between log mel spectrograms of the reconstructed and ground truth waveforms. The configuration of this loss is the same as described in 3.5.

| Ablation on | Decoder dim. | Activation | Multi-period | Single-scale | # of STFT bands | Multi-scale mel. | Latent dim | Quant. method | Quant. dropout | Bitrate (kbps) | Balanced samp. | Mel distance↓ | STFT distance↓ | ViSQOL↑ | SI-SDR↑ | Bitrate efficiency↑ |
|---|---|---|---|---|---|---|---|---|---|---|---|---|---|---|---|---|
| | 1536 | snake | ✓ | ✗ | 5 | ✓ | 8 | Proj. | 1.0 | 8 | ✓ | 1.09 | 1.82 | 3.96 | 9.12 | 99% |
| Architecture | 512 | snake | ✓ | ✗ | 5 | ✓ | 8 | Proj. | 1.0 | 8 | ✓ | 1.11 | 1.83 | 3.91 | 8.72 | 99% |
| | 1024 | snake | ✓ | ✗ | 5 | ✓ | 8 | Proj. | 1.0 | 8 | ✓ | 1.07 | 1.82 | 3.96 | 9.07 | 99% |
| | 1536 | relu | ✓ | ✗ | 5 | ✓ | 8 | Proj. | 1.0 | 8 | ✓ | 1.17 | 1.81 | 3.83 | 6.92 | 99% |
| Discriminator | 1536 | snake | ✗ | ✗ | ✗ | ✓ | 8 | Proj. | 1.0 | 8 | ✓ | 1.13 | 1.92 | 4.12 | 1.07 | 62% |
| | 1536 | snake | ✓ | ✗ | 1 | ✓ | 8 | Proj. | 1.0 | 8 | ✓ | 1.07 | 1.80 | 3.98 | 9.07 | 99% |
| | 1536 | snake | ✗ | ✗ | 5 | ✓ | 8 | Proj. | 1.0 | 8 | ✓ | 1.07 | 1.81 | 3.97 | 9.04 | 99% |
| | 1536 | snake | ✗ | ✓ | 5 | ✓ | 8 | Proj. | 1.0 | 8 | ✓ | 1.08 | 1.82 | 3.95 | 8.51 | 99% |
| Reconstruction loss | 1536 | snake | ✓ | ✗ | 5 | ✗ | 8 | Proj. | 1.0 | 8 | ✓ | 1.10 | 1.87 | 4.01 | 7.68 | 99% |
| Latent dim | 1536 | snake | ✓ | ✗ | 5 | ✓ | 2 | Proj. | 1.0 | 8 | ✓ | 1.44 | 2.08 | 3.65 | 2.22 | 84% |
| | 1536 | snake | ✓ | ✗ | 5 | ✓ | 4 | Proj. | 1.0 | 8 | ✓ | 1.20 | 1.89 | 3.86 | 7.15 | 97% |
| | 1536 | snake | ✓ | ✗ | 5 | ✓ | 32 | Proj. | 1.0 | 8 | ✓ | 1.10 | 1.84 | 3.95 | 9.05 | 98% |
| | 1536 | snake | ✓ | ✗ | 5 | ✓ | 256 | Proj. | 1.0 | 8 | ✓ | 1.31 | 1.97 | 3.79 | 5.09 | 59% |
| Quantization setup | 1536 | snake | ✓ | ✗ | 5 | ✓ | 8 | EMA | 1.0 | 8 | ✓ | 1.11 | 1.84 | 3.94 | 8.33 | 97% |
| | 1536 | snake | ✓ | ✗ | 5 | ✓ | 8 | Proj. | 0.0 | 8 | ✓ | 0.98 | 1.70 | 4.09 | 10.14 | 99% |
| | 1536 | snake | ✓ | ✗ | 5 | ✓ | 8 | Proj. | 0.25 | 8 | ✓ | 0.99 | 1.69 | 4.04 | 10.00 | 99% |
| | 1536 | snake | ✓ | ✗ | 5 | ✓ | 8 | Proj. | 0.5 | 8 | ✓ | 1.01 | 1.75 | 4.03 | 9.74 | 99% |
| | 1536 | snake | ✓ | ✗ | 5 | ✓ | 8 | Proj. | 1.0 | 24 | ✓ | 0.73 | 1.62 | 4.16 | 13.83 | 99% |
| Data | 1536 | snake | ✓ | ✗ | 5 | ✓ | 8 | Proj. | 1.0 | 8 | ✗ | 1.09 | 1.94 | 3.89 | 8.89 | 99% |

Table 2: Results of the ablation study on our proposed codec. The final model is trained with the same configuration as the baseline (top row), but with a quantization dropout of 0.5.

3. STFT distance: distance between log magnitude spectrograms of the reconstructed and ground truth waveforms. We use window lengths $[2048, 512]$. This metric captures the fidelity in higher frequencies better than the mel distance.
4. Scale-invariant source-to-distortion ratio (SI-SDR) [20]: distance between waveforms, similar to signal-to-noise ratio, with modifications so that it is invariant to scale differences. When considered alongside spectral metrics, SI-SDR indicates the quality of the phase reconstruction of the audio.
5. Bitrate efficiency: We calculate bitrate efficiency as the sum of the entropy (in bits) of each codebook when applied on a large test set divided by the number of bits across all codebooks. For efficient bitrate utilization this should tend to 100% and lower percentages indicate that the bitrate is being underutilized.

We also conduct a MUSHRA-inspired listening test, with a hidden reference, but no low-passed anchor. In it each one of ten expert listeners rated 12 randomly selected 10-second samples from our evaluation set, 4 of each domain; speech, music and environmental sounds. We compare our proposed system at 2.67kbps, 5.33kbps and 8kbps to EnCodec at 3kbps, 6kbps and 12kbps.

## 4.5 Ablation study

We conduct a thorough ablation study of our model, varying components of our training recipe and model configuration one-by-one. To compare models, we use the four objective metrics described in Section 4.4. The results of our ablation study can be seen in Table 2.

**Architecture:** We find that varying the decoder dimension has some effect on performance, with smaller models having consistently worse metrics. However, the model with decoder dimension 1024 has similar performance to the baseline, indicating that smaller models can still be competitive. The change with the biggest impact was switching out the *relu* activation for the *snake* activation. This change resulted in much better SI-SDR and other metrics. Similar to the results in BigVGAN [21], we find that the periodic inductive bias of the snake activation is helpful for waveform generation. For our final model, we use the largest decoder dimension (1536), and the snake activation.

**Discriminator:** Next, we removed or changed the discriminators one-by-one, to see their impact on the final result. First, we find that the multi-band STFT discriminator does *not* result in significantly better metrics, except for SI-SDR, where it is slightly better. However, when inspecting spectrograms of generated waveforms, we find that the multi-band discriminator alleviates aliasing of high frequencies. The upsampling layers of the decoder introduce significant aliasing artifacts [29]. The multi-band discriminator is more easily able to detect these aliasing artifacts and give feedback to the generator to remove them. Since aliasing artifacts are very small in terms of magnitude, their effect on our objective metrics is minimal. Thus, we keep the multi-band discriminator.

We find that adversarial losses are critical to both the quality of the output audio, as well as the bitrate efficiency. When training with only reconstruction loss, the bitrate efficiency drops from 99% to 62%, and the SI-SDR drops from 9.12 to 1.07. The other metrics capture spectral distance, and are relatively unaffected. However, the audio from this model has many artifacts, including buzzing, as it has not learned to reconstruct phase. Finally, we found that swapping the multi-period discriminator for the single-scale waveform discriminator proposed in MelGAN [19] resulted in worse SI-SDR. We retain the multi-period discriminator.

**Impact of low-hop reconstruction loss:** We find that low-hop reconstruction is critical to both the waveform loss and the modeling of fast transients and high frequencies. When replaced with a single-scale high-hop mel reconstruction (80 mels, with a window length of 512), we find significantly lower SI-SDR (7.68 from 9.12). Subjectively, we find that this model does much better at capturing certain sorts of sounds, such as cymbal crashes, beeping and alarms, and singing vocals. We retain the multi-scale mel reconstruction loss in our final recipe.

**Latent dimension of codebook:** the latent dimension of the codebook has a significant impact on bitrate efficiency, and consequently the reconstruction quality. If set too low or too high (e.g. 2, 256), quantitative metrics are significantly worse with drastically lowered bitrate efficiency. Lower bitrate efficiency results in effectively lowered bandwidth, which harms the modeling capability of the generator. As the generator is weakened, the discriminator tends to "win", and thus the generator does not learn to generate audio with high audio quality. We find 8 to be optimal for the latent dimension.

**Quantization setup:** we find that using exponential moving average as the codebook learning method, as in EnCodec[8], results in worse metrics especially for SI-SDR. It also results in poorer codebook utilization across all codebooks (Figure 1). When taken with its increased implementation complexity (requiring K-Means initialization and random restarts), we retain our simpler projected lookup method for learning codebooks, along with a commitment loss. Next, we note that the quantization dropout rate has a significant effect on the quantitative metrics. However, as seen in Figure 2, a dropout of 0.0 results in poor reconstruction with fewer codebooks. As this makes usage of the codec challenging for downstream generative modeling tasks, we instead use a dropout rate of 0.5 in our final model. This achieves a good trade-off between audio quality at full bitrate as well as lower bitrates. Finally, we show that we can increase the max bitrate of our model from 8kbps to 24kbps and achieve excellent audio quality, surpassing all other model configurations. However, for our final model, we train at the lower bitrates, in order to push the compression rate as much as possible.

**Balanced data sampling:** When removed, this results in worse metrics across the board. Empirically, we find that without balanced data sampling, the model produces waveforms that have a max frequency of around 18kHz. This corresponds to the max frequency preserved by various audio compression algorithms like MPEG, which make up the vast majority of our datasets. With balanced data sampling, we sample full-band audio from high-quality datasets (e.g. DAPS) just as much as possibly band-limited audio from datasets of unknown quality (e.g. Common Voice). This alleviates the issue, allowing our codec to reconstruct full-band audio, as well as band-limited audio.

## 4.6 Comparison to other methods

We now compare the performance of our final model with competitive baselines: EnCodec [8], Lyra [46], and Opus [37], a popular open-source audio codec. For EnCodec, Lyra, and Opus, we use publicly available open-source implementations provided by the authors. We compare using both objective and subjective evaluations, at varying bitrates. The results are shown in Table 3. We find that the proposed codec out-performs all competing codecs at all bitrates in terms of both objective and subjective metrics, while modeling a much wider bandwidth of 22kHz.

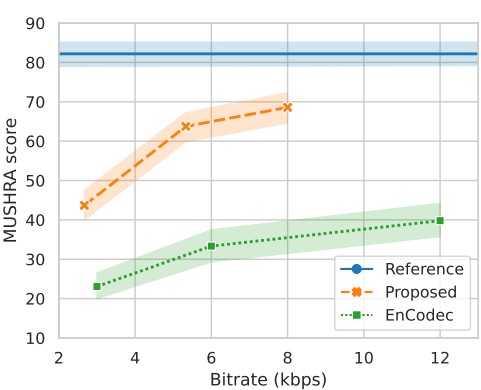

Figure 3: **Listening tests at 44 KHz**: MUSHRA scores, with 95% confidence intervals vs bitrate for EnCodec, our proposed approach, and the reference.

| Codec | Bitrate (kbps) | Bandwidth (kHz) | Mel distance ↓ | STFT distance ↓ | ViSQOL ↑ | SI-SDR ↑ |
|---|---|---|---|---|---|---|
| Proposed | 1.78 | 22.05 | 1.39 | 1.95 | 3.76 | 2.16 |
| | 2.67 | 22.05 | 1.28 | 1.85 | 3.90 | 4.41 |
| | 5.33 | 22.05 | 1.07 | 1.69 | 4.09 | 8.13 |
| | 8 | 22.05 | 0.93 | 1.60 | 4.18 | 10.75 |
| EnCodec | 1.5 | 12 | 2.11 | 4.30 | 2.82 | -0.02 |
| | 3 | 12 | 1.97 | 4.19 | 2.94 | 2.94 |
| | 6 | 12 | 1.83 | 4.10 | 3.05 | 5.99 |
| | 12 | 12 | 1.70 | 4.02 | 3.13 | 8.36 |
| | 24 | 12 | 1.61 | 3.97 | 3.16 | 9.59 |
| Lyra | 9.2 | 8 | 2.71 | 4.86 | 2.19 | -14.52 |
| Opus | 8 | 4 | 3.60 | 5.72 | 2.06 | 5.68 |
| | 14 | 16 | 1.23 | 2.14 | 4.02 | 8.02 |
| | 24 | 16 | 0.88 | 1.90 | 4.15 | 11.65 |

Table 3: Objective evaluation of the proposed codec at varying bitrates, along with results from competing approaches.

In Figure 3, we show the result of our MUSHRA study, which compares EnCodec to our proposed codec at various bitrates. We find that our codec achieves much higher MUSHRA scores than EnCodec at all bitrates. However, even at the highest bitrate, it still falls short of the reference MUSHRA score, indicating that there is room for improvement. We note that the metrics of our final model are still lower than the 24kbps model trained in our ablation study, as can be seen in Table 2. This indicates that the remaining performance gap may be closed by increasing the maximum bitrate.

In Figure 4 and Table 4, we compare our proposed model trained with the same exact configuration as EnCodec (24 KHz sampling rate, 24 kbps bitrate, 320 stride, 32 codebooks of 10 bits each) to existing baselines, in both quantitative and qualitative metrics. In Figure 5, we show qualitative results by sound category.

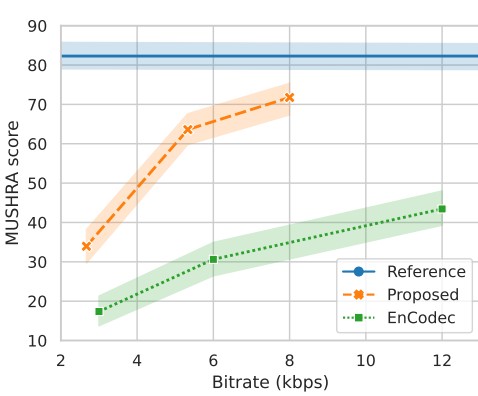

Figure 4: **Listening tests at 24 KHz**: MUSHRA scores with 95% confidence intervals vs bitrate for EnCodec, our proposed approach with the same configuration, and the reference. Here all samples under comparison are resampled to 24 KHz.

| Codec | Bitrate (kbps) | Bandwidth (kHz) | Mel distance ↓ | STFT distance ↓ | ViSQOL ↑ | SI-SDR ↑ |
|---|---|---|---|---|---|---|
| Proposed@24kHz | 1.5 | 12 | 1.48 | 2.24 | 4.04 | 0.32 |
| | 3 | 12 | 1.24 | 2.01 | 4.23 | 4.44 |
| | 6 | 12 | 1.00 | 1.78 | 4.38 | 8.44 |
| | 12 | 12 | 0.74 | 1.54 | 4.51 | 12.51 |
| | 24 | 12 | 0.49 | 1.33 | 4.61 | 16.40 |
| EnCodec | 1.5 | 12 | 1.63 | 2.69 | 3.98 | 0.02 |
| | 3 | 12 | 1.46 | 2.54 | 4.16 | 2.99 |
| | 6 | 12 | 1.30 | 2.39 | 4.30 | 6.06 |
| | 12 | 12 | 1.15 | 2.28 | 4.39 | 8.44 |
| | 24 | 12 | 1.05 | 2.21 | 4.42 | 9.69 |

Table 4: **Encodec Configuration**: Objective evaluation of the proposed model trained with the same configuration as EnCodec at varying bitrates, along with results from EnCodec.

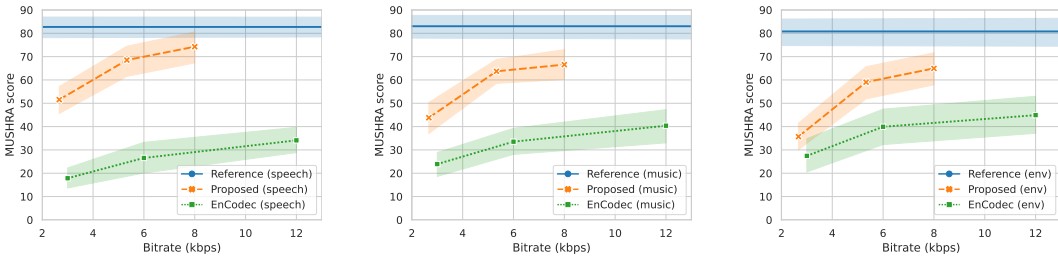

Figure 5: **MUSHRA by category**: MUSHRA scores with 95% confidence intervals vs bitrate for our proposed model, EnCodec and reference.

## 5   Conclusion

We have presented a high-fidelity universal neural audio compression algorithm that achieves remarkable compression rates while maintaining audio quality across various types of audio data. Our method combines the latest advancements in audio generation, vector quantization techniques, and improved adversarial and reconstruction losses. Our extensive evaluation against existing audio compression algorithms demonstrates the superiority of our approach, providing a promising foundation for future high-fidelity audio modeling. With thorough ablations, open-source code, and trained model weights, we aim to contribute a useful centerpiece to the generative audio modeling community.

**Broader impact and limitations:** our model has the capability to make generative modeling of full-band audio much easier to do. While this unlocks many useful applications, such as media editing, text-to-speech synthesis, music synthesis, and more, it can also lead to harmful applications like deepfakes. Care should be taken to avoid these applications. One possibility is to add watermarking and/or train a classifier that can detect whether or not the codec is applied, in order to enable the detection of synthetic media generated based on our codec. Also, our model is not perfect, and still has difficulty reconstructing some challenging audio. By slicing the results by domain we find that, even though the proposed codec outperforms competing approaches across all of the domains, it performs best for speech and has more issues with environmental sounds. Finally, we notice that it does not model some musical instruments perfectly, such as glockenspeil, or synthesizer sounds.

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

## A  Appendix

**Modified codebook learning algorithm**    In our work, we use a modified quantization operation, given by:

$$z_q(x) = W_{\text{out}}e_k, \quad \text{where} \quad k = \arg\min_j ||\ell_2(W_{\text{in}}z_e(x)) - \ell_2(e_j)||_2$$

Here, $W_{\text{in}}$ and $W_{\text{out}}$ are projection matrices, with $W_{\text{in}}$ mapping the encoder's output to an intermediate representation, and $W_{\text{out}}$ mapping this intermediate representation to the quantized representation $z_q(x)$. Specifically,

$$W_{\text{in}} \in \mathbb{R}^{D \times M} \quad \text{and} \quad W_{\text{out}} \in \mathbb{R}^{M \times D}$$

where $D$ is the output dimension of the encoder, and $M$ is the codebook dimension with $M \ll D$.

The vector quantizer loss function is then defined to measure the reconstruction error and is given by:

$$z_{\text{proj}}(x) = W_{\text{in}}\, z_e(x)$$

$$\mathcal{L}_{\text{VQ}} = ||\text{sg}[\ell_2(z_{\text{proj}}(x))] - \ell_2(e_k)||_2^2 + \beta||\ell_2(z_{\text{proj}}(x)) - \text{sg}[\ell_2(e_k)]||_2^2$$

where sg is the stop gradient operator, preventing the back-propagation of gradients through $e_k$, and $\beta$ is a hyperparameter controlling the balance between the two terms in the loss function.

