# OpenReview forum: "High-Fidelity Audio Compression with Improved RVQGAN"
_NeurIPS.cc/2023/Conference — NeurIPS 2023 spotlight_

### Official Review · Reviewer_5w24 · 2023-07-05

**Soundness:** 3 good
**Presentation:** 3 good
**Contribution:** 2 fair
**Rating:** 7
**Confidence:** 5

**Summary:**

In the present paper, the authors introduce RVQGAN, a neural audio codec that uses a convolutional encoder / decoder along with Residual Vector Quantization as a bottleneck, with a multi scale mel reconstruction loss and different adversarial losses.
They show state of the art performance from 3 to 8kbps, compared with the EnCodec model [8].

The key novelties are:
- in each VQ layer, the authors perform the retrieval of the nearest codebook entry into a lower dimension space, and use cosine similarity instead of L2 distance to boost the utilization of the codebooks.
- the authors drop the exponential moving average rule for learning the codebooks.
- the author notice that the original technique from Soundstream [45] to select a varying number of quantizers can hurt the full bandwidth performance, and thus select 50% of the time all the quantizers in RVQ.
- refinement of the losses and adversaries from previous work (in particular using different weights for different frequency bands).
- balancing of the dataset to sample more often fullband audio.

The authors provide extensive ablation studies with objective metrics, and one subjective comparison with EnCodec with various bitrates.

**Strengths:**

- great execution and illustration of the various issues tackled here and the proposed solutions.
- quality of the final model clearly surpasses the existing state of the art.
- detailed ablation study with objective metrics.
- single model for fullband audio over multiple audio domains.

**Weaknesses:**

- incremental improvement over previous work: overall method is coming from [45], adversarial losses are a combination of the one from [45] and [8]. Minor changes to the objective loss compared with [15, 45]. The authors however claim novelty: l.59, "we make impactful design changes [...]: multi scale stft discriminator, multi scale losses".
- some details are unclear to me, in particular, the authors mention they do not use the EMA rule from [9]. How are the codebooks updated then? The authors also mention a low dimension projection, but do not mention when and how it is computed and updated. See questions.
- no ablation with subjective evaluations: could have been interesting to clearly identify where most of the subjective gains is coming from e.g. is it from the quantizer, adversarial losses or dataset balancing?
- seems like the authors do a comparison of a 24kHz baseline model with a 44.1kHz, keeping the ground truth as 44.1kHz, which can have a high impact on subjective and objective metrics independently of the design choices made by the authors. In particular the visqol for Encodec in Table 3 is much lower than reported in [8].

**Questions:**

As mentioned before, I would need more clarification on the exact algorithm used for VQ. Is a PCA computed ? if so how often is it updated (as the encoder output distribution might change). How are the codebooks updated ?

In Section 3.4, the architecture for the multi scale discriminator is missing. Is it the same one as [8] or [45]?

Paragraph starting 193: this insight has been noted and motivated before in [45], [5] and [40], it doesn't seem like the authors bring any new material evidence here?

It would be interesting to see the breakdown of Figure 3 by the category of audio.

A last question would be over learnt snake activation parameters. Do the authors have any insight over the distribution of the learnt $\alpha$? does this vary with the layer? I'm trying to get a better sense of how exactly the model is utilizing this feature.




**Limitations:**

authors properly address societal impact.

---

> ### Author Rebuttal · Authors · 2023-08-04
>
> Thank you very much for you review and constructive criticism of our model. We appreciate the effort in getting to understand the details of our model, and we’ve made our best efforts to answer each of your concerns and questions.
>
> > *incremental improvement over previous work: overall method is coming from [45], adversarial losses are a combination of the one from [45] and [8]. Minor changes to the objective loss compared with [15, 45]. The authors however claim novelty: l.59, "we make impactful design changes [...]: multi scale stft discriminator, multi scale losses"*
> >
>
> We discussed novelty concerns at length in the global rebuttal. We sincerely request the reviewer to review that for additional context.
>
> > *some details are unclear to me, in particular, the authors mention they do not use the EMA rule from [9]. How are the codebooks updated then? The authors also mention a low dimension projection, but do not mention when and how it is computed and updated. See questions*
> >
>
> The residual vector quantization algorithm is an improved version of the one in SoundStream [45], where the codebook learning is modified, inspired by the techniques proposed in Improved VQGAN[43], to encourage uniform codebook usage thereby preventing a “collapse” where many codebook entries are unused.
>
> Specifically, while the standard quantization operation is implemented as follows:
>
> $$
> \displaylines{
> z_q(x) = e_k,\quad \text{where}\quad k = \text{argmin}_j ||z_e(x) - e_j ||_2 \\\ \text{e is the codebook, $z_e$ is the encoder, $z_q$ is the quantizer}
> }
> $$
>
> The modified quantization operation used in our work applies the following equation:
>
> $$
> \displaylines{
> z_q(x) = W_\text{out} e_k,\quad \text{where}\quad k = \text{argmin}_j ||\ell_2(W_\text{in}z_e(x)) - \ell_2(e_j) ||_2  \\\ \text{where $W_\text{in} \in R^{D \times M}$ and $W_\text{out} \in R^{M \times D}$ are projection matrices,} \\\ \text{$D$ is the output dimension of encoder, $M$ is the codebook dimension, $M \ll D$}
> }
> $$
>
> As illustrated in the paper as well as [43], performing codebook lookup in a low-dimensional space leads to improved codebook utilization. In our work, D = 1024 and M = 8.
>
> Also, we noticed that directly updating the codebook using the loss functions in the original VQ-VAE paper instead of EMA is sufficient, simpler to implement and leads to slight performance improvements. Specifically, the loss function is defined as follows:
>
> $$
> \displaylines{
> z_\text{proj}(x) = W_\text{in}\ z_e(x) \\\ \mathcal{L}_\text{VQ} = ||\text{sg}[\ell_2(z_\text{proj}(x))] - \ell_2(e_k) ||_2^2  + \beta \  ||\ell_2(z_\text{proj}(x)) - \text{sg}[\ell_2(e_k)] ||_2^2 \\\ \text{where sg is the stop gradient operator}
> }
> $$
>
> > *seems like the authors do a comparison of a 24kHz baseline model with a 44.1kHz, keeping the ground truth as 44.1kHz, which can have a high impact on subjective and objective metrics independently of the design choices made by the authors. In particular the visqol for Encodec in Table 3 is much lower than reported in [8]*
> >
>
> Thanks for the important feedback. We revisited comparisons against relevant work and discussed it at length in the global rebuttal. Please review that for additional clarifications.
>
> > *In Section 3.4, the architecture for the multi scale discriminator is missing. Is it the same one as [8] or [45]?*
> >
>
> The architecture for the Discriminators is as follows. We use the same Multi-period Discriminator architecture from HifiGAN (cite), and the multi-scale STFT discriminator architecture from UnivNet except with complex STFT as input rather than magnitude spectrograms. Additionally, we do multi-band processing by splitting the spectrogram into sub-bands and using separate discriminator weights for each sub-band, as motivated in our paper (Section 3.4 and Section 4.5).
>
> Also, the code attached to the submission has further exact details on the implementation.
>
> > *Paragraph starting 193: this insight has been noted and motivated before in [45], [5] and [40], it doesn't seem like the authors bring any new material evidence here?*
> >
>
> While this may seem obvious, earlier work such as [45], [5] and [40] haven’t explicitly stated this intuition or shared audio samples with reconstructions from different quantizers. We found this may be interesting to some readers, especially with a practical demo.
>
> Moreover, this intuition is important to understand since prior work such as EnCodec use quantizer dropout only in groups, rather than at each level, and this could impact downstream performance when learning language models on top of the tokens. For instance, we have internally experienced that training language models on top of a codec trained without any quantizer dropout leads to poor audio quality.
>
> > *It would be interesting to see the breakdown of Figure 3 by the category of audio.*
> >
>
> We have attached a pdf to the global rebuttal with the requested breakdown.
>
> > *A last question would be over learnt snake activation parameters. Do the authors have any insight over the distribution of the learnt? does this vary with the layer? I'm trying to get a better sense of how exactly the model is utilizing this feature.*
> >
>
> Thanks for the insightful question. We were just as curious about the distribution of the snake activation parameters, and expected it to correlate with the frequencies being learnt or introduced across the generator. However, we did not find any clear patterns in the parameters in each layer (across different dimensions) or across layers.

---

> > ### Comment · Reviewer_5w24 · 2023-08-11
> > **replying to the authors**
> >
> > I would like to thank the authors for their detailed response. I would kindly ask them to repost the explanation on the updated RVQ rules as it seems the formatting is broken and hence hard to parse.

---

> > > ### Author Response · Authors · 2023-08-11
> > >
> > > Thank you for bringing this to our notice. We have posted an additional comment with the corrected Latex code.

---

> > ### Author Response · Authors · 2023-08-11
> > **Latex correction for correct equation rendering**
> >
> >
> > We present the equations above again with the correct latex code.
> >
> > ---
> > The modified quantization operation equation used in our work:
> > $$
> > z_q(x) = W_\text{out} e_k, \quad \text{where}\quad k = \text{argmin}_j ||\ell\_2(W\_\text{in}z_e(x)) - \ell\_2(e_j) ||_2
> > $$
> >
> > $$
> > \text{$W_\text{in} \in R^{D \times M}$ and $W_\text{out} \in R^{M \times D}$ are projection matrices}\\
> > $$
> >
> > $$
> > \text{$D$ is the output dimension of encoder}
> > $$
> >
> > $$
> > \text{$M$ is the codebook dimension, $M \ll D$}
> > $$
> >
> > ---
> > The vector quantizer loss function:
> > $$
> > z\_\text{proj}(x) = W\_\text{in}\ z_e(x)
> > $$
> >
> > $$
> > \mathcal{L}_\text{VQ} = ||\text{sg}[\ell\_2(z\_\text{proj}(x))] - \ell\_2(e_k) ||_2^2  + \beta ||\ell\_2(z\_\text{proj}(x)) - \text{sg}[\ell\_2(e_k)] ||_2^2 \\ \text{where sg is the stop gradient operator}
> > $$

---

> > > ### Comment · Reviewer_5w24 · 2023-08-14
> > > **continuing the discussion**
> > >
> > > I thank the authors for the updated equations. I would ask the authors to include those equations in the paper in order to make it self contained, especially since the L_VQ loss contains some hyperparameter $\beta$, which as far as I can say are not included in the paper. The authors should specify this value.
> > >
> > > On the other aspects I'm happy with the extra results provided by the authors. On the condition that those are included in the main paper I'm increasing my rating to accept.

---

> > > > ### Author Response · Authors · 2023-08-16
> > > > **Final response**
> > > >
> > > > We thank the reviewer for all the helpful comments. We will definitely update the paper for camera ready version with the updated equations as well as the additional results shared during the rebuttal. We truly believe these updates will significantly strengthen the paper which motivates us to update the final version.

---

### Official Review · Reviewer_ZhX4 · 2023-07-06

**Soundness:** 3 good
**Presentation:** 3 good
**Contribution:** 3 good
**Rating:** 7
**Confidence:** 4

**Summary:**

This paper introduces a novel high-fidelity neural audio compression algorithm that achieves impressive compression ratios while maintaining audio quality. The authors combine advancements in high-fidelity audio generation with improved vector quantization techniques from the image domain, along with enhanced adversarial and reconstruction losses. Their approach achieves a remarkable 90x compression of 44.1 KHz audio into tokens at just 8kbps bandwidth. One of the notable strengths of this work is its universal applicability, as it can compress various audio domains (speech, environment, music) using a single model.

The authors conduct a thorough comparison with competing audio compression algorithms and demonstrate the superior performance of their method. Furthermore, they provide detailed ablations for each design choice, allowing readers to gain insights into the effectiveness of different components. Additionally, the paper offers open-source code and trained model weights, which contribute to the reproducibility of the results.

**Strengths:**

- **Impressive compression performance**: The proposed algorithm achieves a 90x compression ratio for 44.1 KHz audio at just 8kbps bandwidth, demonstrating its effectiveness in reducing data size while preserving audio quality.
- **Novel Method**: The proposed "codebook collapse" and "quantizer dropout" effectively address the issues in lossy audio compression.
- **Universal applicability**: The single model's ability to compress various audio domains makes it highly versatile and applicable to generative modeling of different audio types.
- **Comprehensive evaluation**: The authors compare their method against existing audio compression algorithms, demonstrating its superiority in terms of performance.
- **Thorough ablations**: The paper provides detailed insights into the impact of design choices, allowing readers to understand the effectiveness of different components and their contributions to the overall results.
- **Reproducibility**: The availability of open-source code and trained model weights enhances the reproducibility of the research, enabling other researchers to build upon and validate the findings.

**Weaknesses:**

- The novelty of the proposed model structure is a combination of existing models:
  - factorized codes and L2-normalized codes are from Improved VQGAN image model;
  - Snake activation function from BigVGAN
- This paper presents a strong audio compression technique. However, since the proposed novel points are specifically tailored for a narrow domain, their impact may be limited to the machine learning community and other domains like computer vision/NLP

**Questions:**

- Have you attempted to apply a similar architecture to the vocoder in TTS?
- Which components do you believe can be applied and generalized to other domains or tasks?

**Limitations:**

The authors have adequately addressed the limitations.

---

> ### Author Rebuttal · Authors · 2023-08-04
>
> Thank you very much for your time and feedback. Please find below answers and clarifications to your questions.
>
> > *The novelty of the proposed model structure is a combination of existing models:*
> >
> > - *factorized codes and L2-normalized codes are from Improved VQGAN image model;*
> > - *Snake activation function from BigVGAN*
>
> We have addressed the novelty concern in the global rebuttal. We sincerely request you to review them for additional context.
>
> > *Have you attempted to apply a similar architecture to the vocoder in TTS?*
> >
>
> > *Which components do you believe can be applied and generalized to other domains or tasks?*
> >
>
> While this specific work is focused on an improved technique / algorithm for training neural audio codecs, much of the new recipe is widely applicable to other audio generation tasks such as speech enhancement, source separation, neural vocoding, etc. Internally, we have trained it for the tasks of neural vocoding (trained by removing the encoder and quantization step) and speech enhancement (trained by removing quantization step, while retaining the adversarial losses) and found it to perform as good or better than state of the art models for the respective tasks.
>
> While some of our recipe is borrowed from the image domain, it would also be influential to apply some audio generation techniques to other domains like image/video . For example: periodic activations, residual vector quantization and spectral reconstruction losses haven’t been deeply explored in the image domain and there is interesting scope for future work on this angle.

---

### Official Review · Reviewer_B66v · 2023-07-06

**Soundness:** 4 excellent
**Presentation:** 3 good
**Contribution:** 4 excellent
**Rating:** 7
**Confidence:** 4

**Summary:**

This paper introduces a RVQGAN-based neural audio codec method, demonstrating superior audio reconstruction quality, a high compression rate, and generalization across diverse audio domains. The authors substantiate the significant performance superiority of their model over alternatives through extensive and thorough qualitative and quantitative experiments. They present and validate their technique to fully utilize residual vector quantization, alongside model, discriminator, and loss design choices for enhanced performance.

**Strengths:**

* The paper addresses some of the key challenges in the neural audio codec domain.
* The authors conducted strong and extensive experiments, providing comprehensive results.
* The reference list appears to be thorough and comprehensive.
* The authors support their findings by sharing the developed model, which is beneficial for the research community.


**Weaknesses:**

* The authors derived the proposed methods from existing studies and experimentally validate them in the neural audio codec domain. This approach seems to compromise the scientific novelty of the research.

**Questions:**

* Could the model be applied to downstream applications such as training text-to-speech (TTS) models? Previous works like EnCodec and SoundStream utilized causal architectures to make them suitable for in-context learning or prompting in TTS tasks.

**Limitations:**

The authors have adequately addressed both the limitations of their research and its possible societal impacts.

---

> ### Author Rebuttal · Authors · 2023-08-04
>
> Thank you very much for your time and feedback. Please find below our comments and clarifications.
>
> > *The authors derived the proposed methods from existing studies and experimentally validate them in the neural audio codec domain. This approach seems to compromise the scientific novelty of the research.*
> >
>
> We addressed the novelty concern in the global rebuttal. We sincerely request you to review them for additional context.
>
> > *Could the model be applied to downstream applications such as training text-to-speech (TTS) models? Previous works like EnCodec and SoundStream utilized causal architectures to make them suitable for in-context learning or prompting in TTS tasks.*
> >
>
> We find that causal architectures for the codec aren’t related to downstream applications such as TTS or music modeling. Causal architectures were traditionally required to support streaming applications, which was the primary reason to work on codecs earlier.
>
> We trained a music generative model trained using our codec, which is capable of creating high quality variations of music with different styles of prompting (publication details are withheld for anonymity). Additionally, we internally trained AudioLM-style text-to-speech models on top of the learned tokens and found it to be capable of generating very high quality speech with minimal artifacts.

---

> > ### Comment · Reviewer_B66v · 2023-08-16
> > **Rebuttal Response**
> >
> > I would like to thank the authors for addressing the raised concerns. I wish to further discuss the last query I presented, and if there are any misunderstandings on my part, I would be grateful for any clarifications.
> >
> > When preparing audio data for training LM-like TTS model, it's typical not to differentiate between the audio for the prompt and the subsequent audio for training. Instead, the entire audio is encoded all at once. In such a setup, a non-causal encoder might cause the prompt code to incorporate trailing audio information.
> > For instance, assuming the length of an audio prompt is 3 seconds and the receptive field of the encoder spans 2 seconds, the prompt code derived from a 5-second ground truth audio won't be confined to just the 0 to 3-second audio data. It will also cover the subsequent 3 to 5 seconds, as they fall within the receptive field of the final code.
> >
> > During the inference of an LM-like TTS model trained in this manner, potential issues can emerge:
> > a) If the entire 0 to 5 second ground truth audio is encoded and the first 3/5 of it is selected as the prompt code, it results in a cheating problem. This is because the last code of the prompt will represent not just the audio information from 0 to 3 second but also from the subsequent 3 to 5 second.
> > b) Conversely, if only the first 3 seconds of the ground truth audio are clipped first and then encoded to be used as the prompt code, the encoded code might be interpreted that the audio from 3 to 5 seconds is silent due to the effect of zero-padding in the encoder.
> >
> > On the contrary, encoders with a causal architecture do not present these issues, making them seemingly more suited for providing audio prompts to LM-like TTS models. I am curious to hear the authors' perspective on this matter.

---

> > > ### Author Response · Authors · 2023-08-16
> > > **Discussion on causality of audio codec for training LM-like models**
> > >
> > > Thanks for bringing up this interesting discussion. In this context of our proposed Improved RVQGAN model, we can easily compute the receptive field of the generator architecture, and we find that one frame in the encoded latent space "sees" 7978 samples of audio @ 44.1 KHz which corresponds to 180 ms (90 ms on each side). Our architecture only involves strided convolutions without any self-attention that limits the receptive field of the encoder (arguably, in a good way). Since the receptive field in the non-causal direction is only 90 ms, we believe that it doesn't cause a significant "bleed" of information.
> > >
> > > Moreover, although the receptive fields strongly overlap between subsequent tokens (due to strided convolutions) we believe there's very less incentive for the model to store overlapped information in the discrete latents since the task of heavy compression (~90x) necessitates the model to judiciously use the information bottleneck to store relevant, unique information to reconstruct audio at high fidelity. We empirically find that the latents learn very "local" (patch-wise) information, in agreement with the findings in SoundStream / AudioLM and the interpretation of them as "acoustic tokens". Ex: we found that any point-wise artifact in the input audio (like a short, sudden click) is exactly reproduced in the output audio when passed through the codec.
> > >
> > > As noted earlier, we have also trained LM-style music and speech models on top of the learned acoustic tokens, and they do not exhibit any issues during inference (with prompting), suggesting that bidirectional codecs (with limited receptive field) don't have systemic problems limiting their usage in this respect.
> > >
> > > Note: when prompting we would only encode the prompt audio of 3 seconds, and not the entire audio of 5 seconds followed by clipping. We also did not observe any zero-padding artifacts. We find that the decision of codec causality and the subsequent generative modeling approach don't generally have a strong interaction. Another data point supporting this is in the image domain models such as Parti, Phenaki which also train causal LMs on top of non-causal image tokenizers. However, we feel that this interaction can be better studied thoroughly as future work, and our open-source code can help in this respect.

---

> > > > ### Comment · Reviewer_B66v · 2023-08-17
> > > > **Response to the authors' comment**
> > > >
> > > > I sincerely appreciate the authors for their time and thorough response. I am fully convinced that this research, beyond audio compression, can serve as a robust backbone for various applications, including music generation and text-to-speech. Thank you for the excellent work!

---

### Official Review · Reviewer_d8D3 · 2023-07-07

**Soundness:** 3 good
**Presentation:** 3 good
**Contribution:** 3 good
**Rating:** 7
**Confidence:** 5

**Summary:**

The authors propose a neural audio codec model that demonstrates superior performance compared to previous works, and present experimental results.

**Strengths:**

- The authors appropriately explain the problem they aim to address.
- Their method is adequately described.
- The authors provide a specific implementation, ensuring reproducibility.
- The claims made are reasonable, and the experiments and results support them.
- The authors' various ablation studies can be helpful for future work.


**Weaknesses:**

- For a neural audio codec to be utilized like traditional audio codecs, it should not fail in any patterns. Data-driven neural audio codecs have not been proven to be sufficiently stable from this perspective. Although the authors divided the original dataset into a training set and evaluation set, it is necessary to validate whether the proposed audio codec works well on more diverse and completely different audio data. Additionally, finding failure cases of previous works and comparing them can serve as strong evidence supporting the superiority of the authors' proposed method.

**Questions:**

Based on the MUSHRA scores curves, it appears that higher bitrates yield better scores, and the highest quality that this method can achieve remains unconfirmed. Is there a specific reason for this?

**Limitations:**

Limitations have been well described.

---

> ### Author Rebuttal · Authors · 2023-08-04
>
> Thank you very much for your  time and feedback. Please find our answers below to clarify some details about our model.
>
> > …*it is necessary to validate whether the proposed audio codec works well on more diverse and completely different audio data. Additionally, finding failure cases of previous works and comparing them can serve as strong evidence supporting the superiority of the authors' proposed method.*
> >
>
> To the best of our best knowledge, our model doesn’t have clear failure cases on any audio domain. However, we noticed that some specific sounds like cymbal crashes in music or glockenspiel type sounds aren’t modeled accurately. We leave this to future work to identify why there’s certain limitations on the specific sounds that can be modeled.
>
> Otherwise, we found that our proposed model clearly fixes failure cases in previous works (such as EnCodec). Specifically, EnCodec poorly models background noise (such as room tone), reverb in speech data, but our proposed model almost perfectly reconstructs such cases. This is illustrated in our samples page.
>
> > *Based on the MUSHRA scores curves, it appears that higher bitrates yield better scores, and the highest quality that this method can achieve remains unconfirmed. Is there a specific reason for this?*
> >
>
> We did not test our model at significantly higher bitrates since our goal was to achieve the highest rate of compression possible. However, we additionally trained our model at 16 kbps max bitrate with 44.1 KHz audio and found our metrics to significantly improve over the 8 kbps model (Table 2 in the attached pdf).

---

### Author Rebuttal · Authors · 2023-08-04

The authors of this paper would like to sincerely thank all the reviewers for their time and feedback. Specifically, we thank the reviewers for positively acknowledging:

1) the key challenges in the neural audio codec domain successfully addressed in this work
2) the strong results significantly improving over state of the art
3) comprehensive evaluation and ablation studies that provide insight into all design choices
4) reproducibility of the research through open-sourced code and model weights

We also appreciate and welcome constructive criticism provided for our work and hope to resolve some common concerns noted by the reviewers.

### 1. Addressing novelty concerns

The authors would like to clarify and reiterate that we don’t claim novelty in terms of inventing new techniques, in this paper. However, there is a strong novel contribution of understanding fundamental limitations in existing codec algorithms, and mitigating them with techniques available at our disposal. Each change was strongly motivated by limitations in existing models that have not been addressed till date. Specifically:

1. We found a fundamental limitation in existing audio codecs (such as SoundStream, EnCodec, etc.) that under-utilize their bandwidth due to poor codebook learning (or codebook collapse). This sub-optimal bandwidth usage prevented these models from achieving better quality and higher compression rates which is desirable.
2. We studied the quantizer dropout technique in greater detail and found that it could hurt full bandwidth reconstruction of the model. Addressing this leads to better perceptual reconstruction quality, while still maintaining the benefits of quantizer dropout.
3. While techniques such as periodic (Snake) activations, multi-scale spectral loss, multi-scale STFT discriminator already exist in prior work, there is little consensus in the audio community on the impact of these changes or the motivation behind them. For instance, Snake activation introduced in BigVGAN was not adopted by EnCodec even though it existed earlier. Our work uncovers the theoretical limitations that motivates each of these design changes and also provides thorough ablations for each change. As an additional example, we note in our work that using multi-scale mel reconstruction loss with a lowest hop size of 8 (and filter length 32) leads to a much better modeling of quick transients that exist in audio domains such as music.

We find that scientifically understanding each limitation, and addressing each of them with well-studied and motivated techniques leads to a significant improvement over the state of the art, even leading to a 2x improvement in key metrics such as mel distance (see Table 1 in attached PDF). We request the reviewers to consider the scientific novelty in our work in identifying key limitations of existing models, as well as the disciplined usage of known techniques to mitigate them.

### 2. Revisiting comparisons against relevant work

The reviewers noted that some of our comparisons against baseline models (like EnCodec) are at different sampling rates, which could affect objective and subjective metrics. While this is a valid concern, this was a difficult choice we originally made since downsampling the proposed model would be an unfair comparison. While EnCodec runs natively at 24kHz, by downsampling the output of the proposed model from 44.1kHz to 24kHz we discard all the capacity and bitrate that was allocated to these higher frequencies.

**Listening tests at 24 KHz**: To remove the bandwidth concerns, we downsampled all examples to 24 KHz and redid the subjective listening tests. While this puts our proposed work at a further disadvantage, since it was trained to compress higher bandwidth audio with higher compression factors, we find that our model still significantly outperforms baseline methods (Figure 1 in attached pdf).

**Apples-to-apples comparison with EnCodec**: Moreover, we re-trained our proposed model with the same exact configuration as EnCodec (24 kHz sampling rate, 24 kbps bitrate, 320 stride, 32 codebooks of 10 bits each) to make a thorough apples-to-apples comparison. We have attached quantitative evaluations for the same (Table 1 in attached PDF). In summary, our proposed model significantly improved over EnCodec across all metrics, achieving a mel distance (lower is better) of 0.49 compared to 1.05 of EnCodec.

We believe these updated results should further strengthen the significance of our proposed model. We will add these updated results to the table in the paper to make the comparisons straightforward.

---

### Comment · Area_Chair_ZXb5 · 2023-08-18
**Please read rebuttals**

Dear reviewers ZhX4 and d8D3, if you didn't already, please read the rebuttals ASAP and at least acknowledge them explicitly.

Best,
Area Chair

---

### Decision · Program_Chairs · 2023-09-21

**Decision:**

Accept (spotlight)

**Comment:**

This paper presents an improvement in the discriminator of existing Encodec and Soundstream neural audio codecs. It is particularly effective for HiFi audio with low bitrates (3-8kbps). The reviews all agree that the paper is a solid incremental contribution to a real benchmark, with a paper that describes correctly the method and its advantages. The rebuttal and discussion was productive and the state of the paper is suitable for publication for NeurIPS.